# Dietary Interventions for Night Shift Workers: A Literature Review

**DOI:** 10.3390/nu11102276

**Published:** 2019-09-23

**Authors:** Yan Yin Phoi, Jennifer B. Keogh

**Affiliations:** School of Pharmacy and Medical Sciences, University of South Australia, Adelaide 5001, Australia; yanyin.phoi@unisa.edu.au

**Keywords:** diet, dietary intervention, night shift, shift work, review

## Abstract

(1) Background: Night shift workers have greater risks of developing cardiometabolic diseases compared to day workers due to poor sleep quality and dietary habits, exacerbated by circadian misalignment. Assessing effects of dietary interventions on health outcomes among this group will highlight gaps for future research. (2) Methods: A search of studies was conducted on PubMed, Cochrane Library, Embase, Embase Classic, Ovid Emcare, and Google Scholar, from earliest to June 2019. The population–intervention–comparator–outcomes–study design format determined inclusion criteria. (3) Results: 756 articles were retrieved; five met inclusion and exclusion criteria. Six-hundred-and-seventy night shift workers were from healthcare, industrial, and public safety industries. Dietary interventions included two longer-term nutrition programs and three shorter-term adjustments of meal composition, type, and timing. Outcome measures were varied but included weight and cardiometabolic health measures. Nutrition programs found no weight improvement at both six and 12 months; low-density lipoprotein (LDL)-cholesterol levels decreased at six months. Triglycerides peaked after meals at 7:30 pm; glucose and insulin area under the curve peaked after meals at 11:30 pm. (4) Conclusions: Weight loss was not achieved in the studies reviewed but LDL-cholesterol improved. Future studies should investigate the effects of energy reduction and altering meal timing on cardiometabolic risk factors of night shift workers in randomised controlled trials, while assessing hunger, sleepiness, and performance.

## 1. Introduction

Shift workers are people who work rotating shifts, irregular shifts, evening shifts, afternoon shifts, morning shifts, or split shifts [1]. In Australia, shift workers make up 16% of employees aged 15 years and over, comprising 1.4 million people [1]. Shift workers are known to have elevated cardiometabolic risk factors including higher levels of triglycerides, and lower levels of high-density lipoprotein (HDL)-cholesterol [2]. They also have a greater incidence and risk of chronic disease than the general working population, including metabolic syndrome [3] and obesity [2]. These risks are similar when studies focus on night shift workers [4,5,6,7,8]. Night shift work is linked with an increased risk of diabetes [9], blood pressure [10], breast cancer [11], and heart disease [12]; where weight or waist circumference is adjusted for, the association between night shift work and diabetes is attenuated [9], is inconsistent for blood pressure [6,10], and remains for cancer and heart disease [11,12].

To explain the prevalence of such diseases within the general shift work population, poor sleep quality and low sleep duration have been linked with increased risk of obesity, type 2 diabetes, and atherosclerosis [13,14,15,16]. Several studies of shift workers, not specifically night-shift workers, reveal various dietary trends. Studies that compared total energy intake between shift and non-shift workers do not find significant differences [17,18]. Similarly, while a systematic review of 12 studies by Bonham et al. includes two studies that observed a significantly higher energy intake in shift workers than day workers, their meta-analysis of the 12 studies suggests no difference in energy intake between shift and day workers [19]. A possible explanation for greater weight in shift workers may be partly attributed to decreased energy expenditure; however, this is not reported on in these studies. While energy intake is similar between shift and non-shift workers, food choices and habits are different, such that shift workers consume more snacks [18,20], night-time food [18,20], confectioneries, alcoholic beverages, sugar-sweetened beverages, and have a lower fibre intake [21]. Where total calorie intake is not accounted for, shift workers have more refined sugar, fat [22], and night-time snacks [23], along with irregular meals [22,23]. Furthermore, Wirth et al. found that shift workers have more pro-inflammatory diets than day workers, through the dietary inflammatory index, which assesses an individual’s diet based on various macro- and micronutrient components to determine extent of anti- or pro-inflammatory components of a person’s diet [24].

Night shift workers face disruptions to their circadian rhythm, but there is a lack of direct data investigating the effects of circadian dysregulation, along with factors that influence it in this population. However, various human and animal studies have suggested that the circadian rhythm has a role to play in energy metabolism; at the same time, there are various external factors that feedback to affect circadian regulation. The circadian rhythm lasts about 24 h and may be shifted, or disrupted, by light and nutrient intake [25]. It is coordinated by the suprachiasmatic nucleus (SCN) in the hypothalamus, known as the master circadian clock, which synchronises peripheral clocks [26]. One such peripheral clock is the liver, which affects energy metabolism through the production of regulatory proteins and enzymes that are involved in the synthesis of bile acids, carbohydrate metabolism, and fatty acid metabolism [26]. There is evidence of circadian regulation of glucose, insulin, and appetite modulating factors, ghrelin and leptin [25], disruption of which result in adverse effects on cardiometabolic health [7]. Furthermore, timing and food composition have been shown to affect peripheral oscillators of the circadian clock, such that it uncouples from the SCN [26,27,28]. These studies suggest that night shift workers may face circadian misalignment, and when that is coupled with poor food choices and irregular meals, may further impair the diurnal variation of energy metabolism, compounding the negative effects to their health in a vicious and complex cycle.

In considering appropriate dietary interventions for shift workers, it is also important to cater to the management of fatigue and specific requirements of night work [29], such as alertness and performance. Shift workers must deal with fatigue and mood changes [30,31], which may render them less alert with lower on-the-job concentration and poorer job performances, with limited energy to prepare food [31,32,33]. Therefore, dietary interventions need to be feasible and appropriate to the lifestyle of shift workers to enable compliance.

Previous reviews have looked at the effect of workplace health promotion interventions, comprising of advice on diet and/or physical activity, or environmental manipulation, on health and dietary outcomes of general workers [34,35]. Neil-Sztramko et al. reviewed broad health-related interventions while Lassen et al. appraised interventions promoting healthier food and/or physical activity practices of the shift-work population [36,37]. Because of the heterogeneity of the studies within these two reviews, it was difficult to draw significant conclusions of the effect of dietary manipulation alone on health outcomes. Demou et al. conducted a review on group-based healthy lifestyle interventions in shift workers, some of which included dietary interventions, but they did not all involve night shift workers [38]. To our knowledge, there have not been any reviews of dietary interventions alone on night shift workers.

In the nutritional management of night shift workers, both the timing and content of meals are important in addressing the circadian rhythm of digestive and metabolic processes, and direct physiological effects [29]. The aim of this review is to assess the dietary interventions that have been carried out on night shift workers, along with the resultant impact on their health status. This includes overall health, cardiometabolic diseases, deficits in performance, and wellbeing.

## 2. Materials and Methods

A comprehensive review of the literature was carried out via the electronic databases PubMed, Cochrane Library, Embase, Embase Classic, Ovid Emcare, and Google Scholar, investigating dietary interventions of shift workers, published from earliest to June 2019. Reference lists of reviews and obtained articles were also searched for relevant publications. The search strategy was restricted to the English language and human studies. Our final search strategy was: (“shift work*”[tiab] or “night shift”[tiab]) and (diet*[tiab] or nutrition*[tiab] or food*[tiab]). The * allows for variations in the word and [tiab] identifies keywords in both the title and abstract.

A total of 748 publications were found from the electronic search. After excluding duplicates, 738 articles remained. Hand-searching reference lists identified a further 18 citations, resulting in 756 citations for title screening. After title screening, 102 abstracts were reviewed for eligibility, from which 29 publications were identified for full-text assessment. After full-text screening, 24 papers were excluded, with reasons for exclusion noted. This resulted in five articles for review. Figure 1 represents a Preferred Reporting Items for Systematic Reviews and Meta-Analyses (PRISMA) flow diagram of study selection.

Criteria for study inclusion were developed using the population–intervention–comparator–outcomes–study (PICOS) design format.
-Population: Night shift workers, excluding studies on simulated night shift.-Intervention: Dietary intervention. Studies that had additional components to dietary intervention, such as physical activity or other lifestyle interventions, were included, but the additional components would not be discussed.-Comparator: No comparator.-Outcomes: Overall health including anthropometrics, dietary knowledge and behaviour, cardiometabolic profile, overall health and wellbeing, performance, hunger, and compliance.-Study type: All types, excluding reviews.

## 3. Results

In total, five studies were included in our review [39,40,41,42,43]. Reasons for exclusion included study population, study design, outcome not consistent with the inclusion criteria, and study protocol. A summary of data extracted from original articles is presented in Table 1.

Included studies were three from North America [39,40,41], one from Europe [42], and one from Asia [43]. They were conducted in a wide range of workplace settings including hospitals, industrial plants, and fire departments (Table 1). Three studies were randomized controlled trials [40,41,43] and two were pre–post intervention trials [39,42]. In total, this review consists of 670 participants; the smallest study had six participants [39] while the largest 599 [41]. All study participants worked night shifts. Health and medical conditions of participants were not specified, except for two studies that reported exclusion of participants diagnosed with diabetes or metabolic disease [42]; or had chronic medical conditions or were taking medications affecting glucose, thyroid or mental function [43]. The studies ranged in length from three days to a year.

### 3.1. Outcomes

The studies in this review measured a range of health outcomes following dietary intervention, including anthropometrics, dietary knowledge and behaviour, cardiometabolic profile, overall health and wellbeing, performance, and hunger (Table 1).

#### 3.1.1. Anthropometrics, Dietary Knowledge and Behaviour

Two studies collected anthropometric data such as weight, body mass index (BMI), and body composition, along with dietary behaviour and knowledge [40,41]. Both studies by Elliot et al. were comprehensive nutrition education programs. One was an external pilot study of the other; the former being six-months long [40] and the other being a year in duration [41]. Thus, both these studies have similar study protocols, with details highlighting differences in Table 1. Intervention was either team-based nutrition education or one-on-one motivational interviewing sessions, with both intervention groups receiving nutrition, exercise, and health guidebooks for knowledge reinforcement. Both the studies’ control groups were provided baseline health results and asked to modify their lifestyle through their own initiative.

The pilot study of six months found no significant difference in weight, BMI, and body composition compared to baseline or control [40], while the year-long study found an increase in weight and BMI across all groups, with significantly less weight gain in both intervention groups compared to control [41].

In both the studies, only the team-based intervention led to a significant increase in social support and cohesion [40,41]. In the year-long study, dietary understanding significantly improved with team-based education [41] while in the six-month study, personal self-monitoring of diet significantly improved with one-on-one motivational interviewing [40].

In the year-long study, participants completed questionnaires that measured “healthy dietary behaviour” and there was a significant improvement in both team-based and motivational interviewing groups [40,41]. However, narrowing in on particular components of a healthy diet showed inconsistent results. While there was a similar significant increase in fruit and vegetable intake in both groups, there was no decrease in percentage of calories from fat in either group [41]. Conversely, in the six-month study, neither group had a significant difference in fruit and vegetable intake, while there was a decrease in behaviours related to high fat intake through avoidance, replacement, or substitution of high fat foods in the motivational interviewing group but not the team-based group [40].

#### 3.1.2. Cardiometabolic Profiles

Cardiovascular-related outcomes were measured in two studies, including low-density lipoprotein (LDL)-cholesterol, HDL-cholesterol [40], and triglycerides [40,42]. However, the two studies were very different and results cannot be compared. One was a six-month pilot of a comprehensive nutrition education program [40] while the other was a short three-week pre–post intervention study of identical meals consumed at three different time points [42]. The six-month pilot study found significant improvements in LDL in both team-based and motivational interviewing interventions compared to control, with no differences in HDL-cholesterol and triglycerides [40]. On the other hand, Knutsson et al. found a significant increase in triglycerides occurred when a 440 calorie meal was consumed at 7:30 pm compared to 11:30 pm and 3:30 am [42].

Two studies reported on glucose and insulin levels [42,43]. Paz et al. found no significant difference between a high protein, high carbohydrate, and a regular control diet on glucose and insulin levels [43], while Knutsson et al. found that a 440 calorie meal at 11:30 pm led to significant postprandial peaks in glucose at 6 mmol/L and insulin area under the curve (AUC) at 4535 IU/mL compared to meals at 7:30 pm or 3:30 am [42]. Paz and colleagues’ regular control diet was similar in macronutrient composition to Knutsson and colleagues’ test meal; the former at 27% fat, 55% carbohydrate, and 18% protein, while the latter at 33% fat, 51% carbohydrate, and 16% protein. However, the two studies cannot be compared as Paz and colleagues’ study received 1200 calories that they could consume between noon and 4 am, without specific meal timings, while Knutsson et al. provided 440 calorie meals at three specific time points. Blood glucose and insulin concentrations were also drawn at different time points relative to food intake; Paz et al. collected blood samples at 6 am, 2 h before food intake was disallowed while Knutsson et al. collected blood samples at baseline and at 30-min intervals up to 240 min post-meal.

#### 3.1.3. Overall Health, Wellbeing, Performance, and Hunger

Two studies measured emotional wellbeing as part of a 116-item questionnaire within the comprehensive nutrition education programs by Elliot et al. [40,41]. With a one year intervention, there were significant improvements in wellbeing in both team-based and motivational interviewing groups compared to control [41], while a six-month intervention significantly decreased feelings of depression in the motivational interviewing group [40].

Two studies measured sleepiness; both intervened by altering meal composition [39,43]. They used the Stamford Sleepiness Scale (SSS) [39,43] and a trail-making test of visual vigilance and mental alertness [43]. No significant difference in sleepiness was found whether participants were provided with a medium-fat/medium-carbohydrate diet [39], or a high protein or high carbohydrate diet against baseline and control, respectively [43]. These two studies also measured variations of performance, using the Halstaed–Reitan test [43] and a Paced Auditory Serial Addition Test (PASAT) cognitive test [39]. Paz et al. did not find a significant difference in alertness when high protein or high carbohydrate diets were provided compared to a regular control diet [43]. However, a significant improvement in cognitive scores occurred with a medium-fat/medium-carbohydrate diet compared to baseline in Love and colleagues’ study [39]. Paz et al. did not adjust for any variables [43] while Love et al. adjusted for sleep patterns, age, BMI, length of shift work employment, commuting time, and exercise hours [39]. Although both these studies altered meal composition, their results cannot be compared due to differences in study protocol. Love and colleagues’ intervention compared a medium-fat/medium-carbohydrate meal against regular baseline shift conditions at midnight for two night shifts each, one week apart [39]. On the other hand, Paz and colleagues’ intervention compared a high protein or high carbohydrate diet against a regular control diet from noon onwards, on three separate occasions with unspecified gaps in between each test diet [43]. Details of specific dietary compositions are in Table 1. Lastly, they were both small studies; Paz et al. had 21 participants [43], while Love et al. six participants [39]. 

Love et al. was the only study that looked at hunger ratings [39,44]. Replacement of regular meals during shift hours with a medium-fat/medium-carbohydrate meal did not change hunger ratings despite test calorie intake being lower by 27% than baseline days [39].

### 3.2. Compliance and Adverse Events

Two studies reported compliance; they were Elliot and colleagues’ studies including the comprehensive nutrition education programs [40,41]. The year-long study found that 73% of participants attended ≥7 team sessions while 34% attended all team sessions; participants in the motivational interviewing group averaged 4.4 ± 1.5 visits, which is more than the minimum of four visits scheduled [41]. Furthermore, usage of information guide was recorded and there was no significant difference between intervention groups. In the six-month pilot study, attendance was not reported but participants’ time spent at group and individual sessions were approximately 120% and 71% of the time allocated, respectively [40]. Adverse events were not reported in any of the studies.

Attrition rates ranged from 0% to 20%. Attrition rates were 0% in the three short-term studies where test diets were provided on not more than three days [39,42,43]. In the remaining two comprehensive nutrition education programs, attrition rates were 0% in the six-month pilot [40], and 20% in the one-year study [41].

## 4. Discussion

This is the first literature review to focus on dietary interventions exclusively among night shift workers. Overall, clear conclusions could not be drawn due to the small number of studies and varying study protocols. However, the studies produced some observations. Comprehensive nutrition education programs did not result in weight loss, but led to a decrease in LDL-cholesterol at six months. While team-based intervention led to improved social cohesion it did not translate to dietary behaviour change. In terms of meal-time food intake, consuming a meal at 11:30 pm led to peak glucose and insulin AUC levels while a meal at 7:30 pm led to peak triglyceride levels. Lastly, a medium-fat/medium-carbohydrate meal improved performance.

In this review, Elliot and colleagues’ 2004 and 2007 studies suggested that team-based interventions encourage social support and cohesion. As the authors described, the group setting encouraged discussion, friendly competition, and camaraderie. However, whether it translates to significant improvements in dietary understanding; dietary behaviour such as fat, fruit, and vegetable intake; or BMI; compared to motivational interviewing or control remains unclear in the studies. The results from these two studies elicits the question of the effectiveness of dietary intervention with these studies’ intervention methods. While participants in both the intervention groups demonstrated various improvements in dietary habits and knowledge, the translation to meaningful changes in health outcomes such as weight and cardiovascular markers such as lipid profile at one year and beyond was not evident. However, the lack of consistent results could be attributed to the studies’ design. While both the studies’ interventions included advice or motivational interviewing on nutrition and health, participants set their own lifestyle goals to work towards during the course of the study. They also did not receive dietitian support, and there was no mention of calorie deficit for weight loss. Therefore, the goals for each participant may not be exactly in line with study outcomes measured, resulting in a dilution of effect size.

Furthermore, both the studies found an increase in weight in both intervention groups after one year that was not observed in the six-month pilot study. This highlights the issue of weight maintenance in the long term. Other general workplace lifestyle intervention studies that initially found significant improvements in weight outcomes after six months of intervention have also found a loss of significance at one-year follow-up [45,46]. A recent review further concludes that good quality studies on the long-term effectiveness of work-based lifestyle interventions on BMI were lacking [47]. This generates the question of long-term sustainability of such comprehensive programs on health outcomes, especially when resources are removed.

The two comprehensive programs by Elliot et al. focused on nutrition education and included group support, behavioural counselling, and informative materials. This has similarities in intervention style and outcomes to other worksite health programs carried out on non-shift workers [34,48]. However, such interventions and outcomes are generic and do not address the specific work demands or health issues of night shift workers. A qualitative review of train drivers revealed that organisational and social barriers such as resource limitation, poor roster design, and sleepiness-induced fatigue prevented time and energy allocation to adopting healthy lifestyles [49]. Even where workplace health promotion programs were available, nurses cited staff shortages, fatigue, and lack of time as reasons for non-attendance [50]. Therefore, shift work interventions should address work organisation, and account for cultural, environmental, and social factors [51]. General worksite health interventions cannot simply be applied to this population.

Various animal and human mechanistic studies demonstrate potential dietary interventions that could support night shift workers. In mice, food intake during the sleep phase altered circadian clock and metabolic genes, and led to greater weight gain [52]. Furthermore, when mice were given a high-fat, high-calorie diet instead of regular chow in free-running un-entrained conditions, there were alterations in circadian clock gene expression involved in fuel utilization in the hypothalamus, adipose tissue, and liver, independent of body weight [53]. Interestingly, mice limited to food intake of 8 h a day during natural nocturnal times compared to mice fed an equivalent high fat diet ad libitum had improved catabolic and anabolic pathway function and circadian clock oscillation, which optimised nutrient utilisation and energy expenditure, thus preventing glucose intolerance, hyperinsulinemia, fatty liver, inflammation, and obesity [54]. Moving on to human studies of simulated night work, Ribeiro and colleagues observed increased postprandial triacylglycerol levels when a main meal of 796 calories was provided at a body clock time of 10:30 pm [55]. However, in another simulated shift work study, Centofani et al. found that a large snack of 502 calories at midnight led to glucose impairment while a small snack of 201 calories did not [56]. These studies lay the foundation and suggest that avoidance of night-time food intake, or adjustment of meal composition or portion can affect circadian clock regulation and may prevent cardiometabolic disturbances. Yet, such study designs have not been conducted in night shift workers.

In this review, only Knutsson et al. assessed the effect of meal timings on cardiometabolic outcomes such as glucose, insulin, and triglyceride levels [42]. By being conducted on actual shift workers, their findings support the notion of night-time food intake impairing glycaemic profile but not triglyceride levels. Yet, these results from a single non-randomised, non-controlled trial does not provide sufficient evidence of the negative effects of late-night meals on cardiometabolic profiles. Mice and human simulated shift work studies have shown that limiting meal timings hold much potential; this must be translated in more studies of night shift workers. Furthermore, they need to move beyond short-term mechanistic studies towards randomised controlled trials of longer durations.

Love et al. and Paz et al. altered meal composition. However, between the two studies, the timing that study meals were provided and the duration between baseline/control and test periods were different, rendering comparisons between the studies difficult. The foods we eat affect leptin, ghrelin, glucose, fat, and insulin levels, which are shown to act on signals in various pathways to affect the circadian rhythm and thereby circadian regulation of energy metabolism [57]. Therefore, an optimal meal composition may have potential in achieving favourable outcomes on cardiometabolic profile through minimising the misalignment of circadian regulation in night shift work populations. Unfortunately, these two studies did not have improvements in cardiometabolic health as objectives, and were not of a long duration, both of which would have been results of interest. Another point of consideration is that all foods affect glucose, fat, and insulin levels. Therefore, minimising portion sizes of food consumed at night may reduce surges in these parameters, thus ameliorating cardiometabolic and circadian disturbances. These form potential areas for future investigation.

As important as intervention diets in studies of meal timings and composition are appropriate outcome measures. Only Knutsson et al. collected comprehensive glucose, insulin, and triglyceride levels in a bid to understand the effects of meal timings on diabetes and cardiovascular health. Otherwise, essential outcome measures like weight, waist circumference, and lipid profiles were only measured in the longer-term comprehensive nutrition education program by Elliot and colleagues. Love et al. and Paz and colleagues’ objectives were only of cognition, performance, and sleepiness, which acceptably, hold some relevance. This is because shift workers are expected to stay awake and function at work during night hours, which is usually for rest. Nevertheless, to generate completeness in outcome measures relevant to this population, future studies must collect a mixture of anthropometric, cardiometabolic, and circadian outcomes, along with measures of sleepiness and performance at work to assess feasibility.

The three short-term studies investigating meal composition and timing in this review have 0% attrition rates. They had no incentives for participation, but provided test meals, which encourages compliance. This shows the potential ease in conducting future studies that test mediating mechanisms between diet components and health outcomes in night shift workers. Furthermore, as the goal is to investigate the effects of various dietary aspects on cardiometabolic and circadian health of night shift workers, interventions should happen during working hours. This prevents additional demands to be placed on the time or energy of night shift workers, thus hypothetically supporting recruitment and compliance in future studies of this population.

There are several limitations to this literature review. The studies may not have been adequately powered to detect between-group differences, since they did not report power calculations. Coupled with the heterogeneity of intervention types and outcomes, it was difficult to determine the effectiveness of specific interventions on targeted health outcomes, let alone the strength of effect. Similarly, there were insufficient studies of similar protocols for comparison of intervention effectiveness or outcome measures. However, given the limited number of diet studies on this population, this review provides an overview of the dietary interventions attempted on night shift workers and more importantly, highlights areas for further research to improve their specific health issues.

## 5. Conclusions

This review highlights that few studies have been conducted on dietary interventions exclusively among night shift workers. Comprehensive nutrition education programs in this review did not achieve weight loss, although LDL-cholesterol levels decreased after six months. Although team-based intervention encouraged social cohesion, clear dietary behaviour change was not evident. Short-term studies that adjusted meal composition and timing found improved performance with a medium-fat/medium-carbohydrate meal, while meals at 11:30 pm led to peak glucose and insulin AUC levels and meals at 7:30 pm led to peak triglyceride levels. Nevertheless, the studies in this review varied in study design, intervention, and outcome measures, rendering firm conclusions difficult. Furthermore, few studies examined outcome measures pertinent to the health of night shift workers, such as anthropometrics, cardiometabolic profiles or circadian rhythm. Future studies should investigate if limiting food intake to daylight or restricted hours, optimising meal composition, or small snacks instead of main meals at night minimises exacerbation of circadian disruption and downstream effects on cardiometabolic profiles. In addition, effects on hunger, sleepiness, and performance on the job should also be assessed to determine appropriateness of intervention in this population. Only when beneficial dietary aspects and feasibility have been established in larger and longer-term randomised controlled trials of night shift workers can clinical recommendations and policy development be made to improve the health of the night shift work population.

## Figures and Tables

**Figure 1 nutrients-11-02276-f001:**
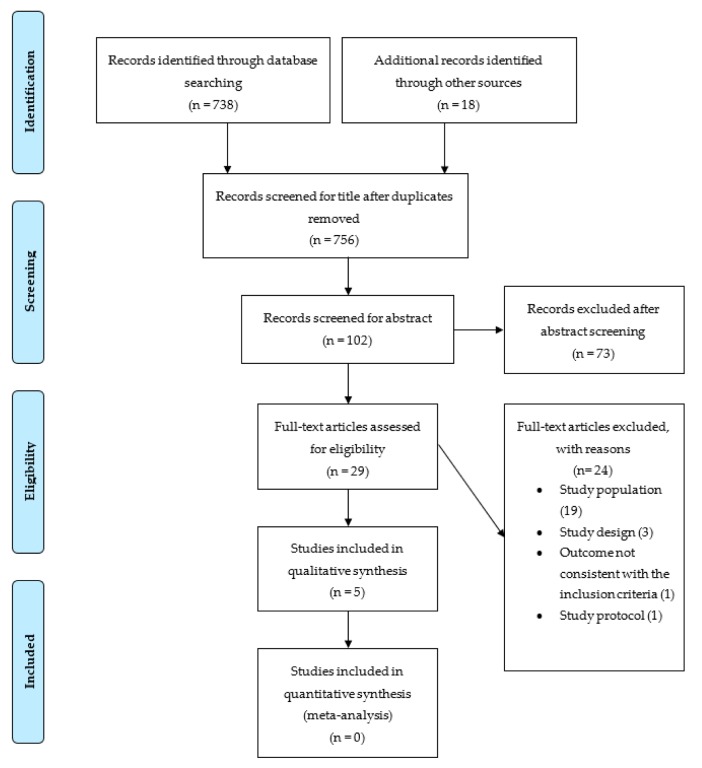
Flow chart of the selection process of included studies.

**Table 1 nutrients-11-02276-t001:** Descriptive table of included studies.

Author (Year), Study Type	Participants	Start (n), Night Shift Workers (n/% of Study Population)	End (n), Night Shift Workers (n/% of Study Population), Attrition (%), Reasons for Attrition	Intervention	Outcome Measures	Results	Compliance of Completers, Adverse Events
Love et al., 2005 [39], two-week pre–post intervention trial; two baseline shift days in 1st week and two test shift days in 2nd week.	Country: Canada Gender: women (*n* = 2) male (*n* = 4) Occupation: industrial plant Age: 19–44 years	Total: 6 Shift: 100%	Total: 6 Shift: 100% Attrition: 0%	Provision of test meal at 12:00 am: Medium-fat, medium-carbohydrate test meal (706.9 kcal, 12% protein, 46% carbohydrate, 42% fat) at 12:00 am consisting of a milk drink and cheese sandwich. Optional addition of snack bar (340 kcal, 18% protein, 53% carbohydrate, 29% fat) 2.5 h after meal to meet differences in energy requirements.	Food intake Paced Auditory Serial Addition Test (PASAT): cognitive test Stanford Sleepiness Scale (SSS): measure of alertness Level of fullness/hunger on a ten-point continuum	Compared to baseline PASAT scores improved for the 1.6-s test. * % of carbohydrates from total energy decreased while % of fat from total energy increased. *	Compliance: not reported. Adverse events: not reported.
Paz et al., 1997 [43], three-days three-armed randomised controlled trial; cross-over design of three diet types on three occasions.	Country: Israel Gender: women (*n* = 12) male (*n* = 9) Occupation: hospital ward and X-ray department staff Age: mean (SD): females 28.9 (3.2) years, males 34.4 (8.2) years	Total: 21 Shift: 100%	Total: 21 Shift: 100% Attrition: 0%	Provision of three diets at 12:00 pm on the day prior to night shift, on three separate occasions, given in random order: High carbohydrate: 4% protein, 70% carbohydrate, 26% fat. Carbohydrate:protein ratio of 17.5. High protein: 52% protein, 20% carbohydrate, 28% fat. Carbohydrate:protein ratio of 0.38. Regular control: 18% protein, 55% carbohydrate, 27% fat. Carbohydrate:protein ratio of 3.1.	Glucose Insulin Amino acids Halstaed–Reitan tests: feelings, stress, and performance SSS: measure of alertness Trail-making tests: visual vigilance and mental alertness	Comparison between groups Within the high protein group, higher glucose and insulin concentrations associated with improved psychometric results. * Within the high carbohydrate group, higher glucose concentrations were associated with more tiredness. *	Compliance: not reported. Adverse events: not reported.
Knutsson et al., 2002 [42], three-weeks pre–post intervention trial: three days of standardised meals followed by two days of three test meals each, flanked by a week of non-intervention on each end.	Country: Sweden Gender: women (*n* = 11) male (*n* = 0) Occupation: nurses Age: 33–53 years	Total: 11 Shift: 100%	Total: 11 Shift: 100% Attrition: 0%	Provision of: Three days of standardised meals prior to test meal (2200 calories; 31% fat, 54% carbohydrate, 16% protein). Identical test meals for two days on night shifts (440 calories; 33% fat, 51% carbohydrate, 16% protein) at 7:30 pm, 11:30 pm, and 3:30 am.	Glucose Insulin Triglycerides Area under the curve (AUC) for glucose, insulin, and triglycerides	Comparison between timings Highest postprandial glucose at 11:30 pm. * Highest fasting insulin and total insulin AUC at 11:30 pm. * Highest postprandial triglycerides at 7:30 pm. *	Compliance: not reported. Adverse events: not reported.
Elliot et al., 2004 [40], six-month pilot three-armed randomised controlled trial	Country: United States Gender: not stated. Occupation: firefighters Age: mean (SD): team-based 48.3 (4.8) years, individual-oriented 40.5 (7.2) years, control 44 (5.8) years.	Total: 33 Control: 11 Team-based: 12 Individual-oriented: 10 Shift: 100%	Total: 33 Control: 11 Team-based: 12 Individual-oriented: 10 Shift: 100% Attrition: 0%	Team-based: 10 × 45-min peer-taught sessions following scripted manual and workbooks. Five weekly sessions followed by five meetings spaced over 20 study weeks. First session: Provided test results and dietary finding from baseline assessments. Team discussions identified goals and attainment methods. Subsequent sessions: addressed topics related to nutrition and exercise. Non-session weeks: Activities reinforcing PHLAME’s lifestyle objectives. Friendly competition and peer pressure encouraged. Firefighters’ Health and Fitness Guide: information about nutrition, physical activity, and lifestyle diseases. Motivational interviewing: 4 × 60-min one-on-one sessions with trained health counsellor (motivational interviewing), followed by up to 4.5 h of extra phone/in-person contacts. One 15-min meeting with physician about test results. Firefighters’ Health and Fitness Guide. Control: Received baseline test results with brief explanations and list of normal values.	Weight, height BMI Body composition Lipid panel: total cholesterol, high density lipoprotein (HDL), low density lipoprotein (LDL) 116-item questionnaire: knowledge, behaviours and beliefs about nutrition, exercise, body weight, and overall health. Dietary habits indexed using standard assessment instruments for fruits, vegetables, and fat intake	Intervention vs. control Both team-based and motivational interviewing led to reduced LDL. * Motivational interviewing led to decreased behaviours related to higher fat intake, increased tracking of personal eating habits, fewer feelings of depression. * Team-based led to increase in personal exercise practices, healthy dietary and physical activity habits. *	Compliance: Team-based: 9 h out of 7.5 h per firefighter. Individual oriented: 6 h out of 8.5 h per firefighter. Adverse events: not reported.
Elliot et al., 2007 [41], one-year three-armed randomised controlled trial	Country: United States Gender: women (*n* = 20) male (*n* = 579) Occupation: firefighters Age: 20–60 years	Total: 599 Control: 163 Team curriculum: 234 Motivational interviewing: 202 Shift: 100%	Total: 480 Control: 129 Team curriculum: 186 Motivational interviewing: 165 Shift: 100% Attrition: 20% Reasons for attrition: Left employment or medical leave (50) Study withdrawal (60) Transferred to team shift (9)	Team curriculum: 11 × 45-min peer-taught sessions following scripted material. Clusters of three, two, three, and three weekly sessions, with final session approximately three months prior to follow-up testing. First session: Provided test results and dietary finding from baseline assessments. Team discussions identified goals and attainment methods. Subsequent sessions: activities about nutrition, physical activity, and energy balance. Non-session weeks had longitudinal bridging activities. Firefighters’ Health and Fitness Guide. Motivational interviewing: 4 × one-on-one counselling with motivational interviewer. Option of up to 5 h of extra in person/phone contact. At six and 10 months, reached by phone and offered extra meetings. Firefighters’ Health and Fitness Guide. Control: Received baseline test results with brief explanations and list of normal values.	Weight, height BMI Body composition 116-item questionnaire	Compared to baseline: Both groups had increased weight and BMI, but intervention group had significantly less weight gain than the control group. * Team curriculum led to increased daily servings of fruits and vegetables, healthy dietary behaviour, dietary understanding, positive dietary social support, and overall well-being. * Motivational interviewing led to increased daily servings of fruits and vegetables, healthy dietary behaviour, and overall well-being. *	Compliance: TC: 73% of attended ≥7 team sessions, 34% attended all team sessions. MI: Participants each had an average of 4.4 ± 1.5 interactions out of a minimum of four sessions. No difference in usage of Fire Fighter Guide; mean of 3–3.3 between MI and team groups, respectively (0 = no usage, 6 = all usage) Adverse events: not reported.

* statistically significant. If outcome measures are not listed in results, no statistically significant difference was found.

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
