# Peer review of "Dietary Interventions for Night Shift Workers: A Literature Review"

_nutrients, 2019, doi:10.3390/nu11102276_

Round 1

Reviewer 1 Report

This article is a review on the effect of dietary interventions for night shift workers. As the authors point out, the scarcity of studies and the lack of homogeneity between them are striking. From my point of view, this finding itself is of interest. In my opinion, the article is well written and comments on the results and limitations of the different studies.
Table 1 covers a lot of information in a small space and I have found it difficult to read and assimilate the content. 

Author Response

Comment 1:

Table 1 covers a lot of information in a small space and I have found it difficult to read and assimilate the content. 

Answer 1:

Thank you for your feedback. The information in Table 1 has been condensed to address the reviewer’s concern while retaining information the reader may find useful.

Reviewer 2 Report

Shift work, especially night shift work has been associated with negative health effects and is a cause of concerns due to the large percentage of workforce working on night shifts. In this article, the authors focused on the benefits of dietary intervention on the health and wellbeing of night shift workers and highlighted the gaps for future research.

Comment1: Lines mentioned below (Line 43 - 48) are not clear. Authors should rewrite for clarification. On line 43, authors mentioned that the total energy intake between shift and non-shift workers was not found to be different, but on the next few lines mentioned studies that show greater energy intake in shift workers. Additionally, both the reviews mentioned (by Bonham et al., and Lowden et al.) reported 2 studies observing higher energy intake in shift workers. It is not clear if both the reviews included the same studies or not.

Line 43 – 48: Total energy intake between shift and non-shift workers was not found to be different [17-20]. A systematic review by Bonham et al. found no difference in energy intake between shift workers and day workers, with only 2 studies observing a significantly higher energy intake in shift workers than day workers [21]. Similarly, in another review by Lowden et al., 8 studies showed no difference in energy intake between morning, afternoon, and night shift workers, while 2 found greater energy intake in night shift workers. [22].

Comment 2: Figure 1 – Identification section, Records identified through database searching should be 738 instead of 748 as 738 is the actual number after removal of duplicates. This will also justify having 756 (738+18) total records screened for the title. Figure 1 also has numbers outside the boxes (134, 138, 139 etc.). It is not clear what those numbers mean.

Comment 3:   Table 1, column 3 and 4 headings should include “night shift workers” instead of just “shift workers” to avoid any confusions.

Minor comments:

Line 224: “Two studies that measured sleepiness both intervened by altering meal composition [39,43].”

-          Remove “that”

Sentences require restructuring for clarifications:

Line 61:  “However, various human and animal studies have suggested that the circadian rhythm has a role to play in energy metabolism, with certain factors may feed back to affect its regulation.”

Line 93: “The aim of this review is to assess the dietary interventions that have been 94 carried out on night shift workers and impacts on their health status.”

Author Response

Comment 1:

Lines mentioned below (Line 43 - 48) are not clear. Authors should rewrite for clarification. On line 43, authors mentioned that the total energy intake between shift and non-shift workers was not found to be different, but on the next few lines mentioned studies that show greater energy intake in shift workers. Additionally, both the reviews mentioned (by Bonham et al., and Lowden et al.) reported 2 studies observing higher energy intake in shift workers. It is not clear if both the reviews included the same studies or not.

Line 43 – 48: Total energy intake between shift and non-shift workers was not found to be different [17-20]. A systematic review by Bonham et al. found no difference in energy intake between shift workers and day workers, with only 2 studies observing a significantly higher energy intake in shift workers than day workers [21]. Similarly, in another review by Lowden et al., 8 studies showed no difference in energy intake between morning, afternoon, and night shift workers, while 2 found greater energy intake in night shift workers. [22].

Answer 1:

Thank you for your feedback. These sentences have been rewritten to clarify the findings from Bonham et al.’s study. Lowden et al.’s review was removed as it included some studies that were similar to Bonham et al.’s. These changes are from line 43-47, please see below.

“Studies that compared total energy intake between shift and non-shift workers did not find significant differences [17,18]. Similarly, while a systematic review of 12 studies by Bonham et al. included 2 studies that observed a significantly higher energy intake in shift workers than day workers, their meta-analysis of the 12 studies suggested no difference in energy intake between shift and day workers [19].”

Comment 2:

Figure 1 – Identification section, Records identified through database searching should be 738 instead of 748 as 738 is the actual number after removal of duplicates. This will also justify having 756 (738+18) total records screened for the title. Figure 1 also has numbers outside the boxes (134, 138, 139 etc.). It is not clear what those numbers mean.

Answer 2:

In figure 1, the number of records identified through database searching has been changed to 738. We do not see the numbers outside the boxes in the copy of the manuscript that we downloaded, perhaps they are the line numbers that could be seen in your copy due to formatting issues.

Comment 3:   Table 1, column 3 and 4 headings should include “night shift workers” instead of just “shift workers” to avoid any confusions.

Answer 3:

In Table 1, the amendments to column 3 and 4 headings have been changed to “night shift workers”.

Minor comment 4:

Line 224: “Two studies that measured sleepiness both intervened by altering meal composition [39,43].”

-          Remove “that”

Answer 4:

“That” has been removed

Minor comment 5:

Sentences require restructuring for clarifications:

Line 61: “However, various human and animal studies have suggested that the circadian rhythm has a role to play in energy metabolism, with certain factors may feed back to affect its regulation.”

Line 93: “The aim of this review is to assess the dietary interventions that have been 94 carried out on night shift workers and impacts on their health status.”

Answer 5:

The sentences have been restructured and clarified. The changes may be found at lines 64-65 and line 97. Please see below.

64-65: “However, various human and animal studies have suggested that the circadian rhythm has a role to play in energy metabolism; at the same time, there are various external factors that feedback to affect circadian regulation.”

96-97: “The aim of this review is to assess the dietary interventions that have been carried out on night shift workers, along with the resultant impact on their health status.”

This manuscript is a resubmission of an earlier submission. The following is a list of the peer review reports and author responses from that submission.

Round 1

Reviewer 1 Report

The negative effects of shift work and night shift work are well known and have become a cause of concern because of the large percentage of workers with this type of schedule. One of the studied factors is the effect of meals in terms of composition and timing. Being a modifiable factor, it is of especial interest. This paper is a review of the studies that have evaluated the effect of dietary interventions on the health of night shift workers. 

Major Comments: 

“Material and Methods” Section 

Taking into account the selected articles, it is not clear to me whether the data the authors analyze are specific to night shift workers as the authors claim. They include 5 papers where only a certain percentage of the study population is specifically night shift workers. In my opinion, these studies should have subgroup analysis (not specified) to be suitable for the inclusion in the review. Furthermore, one study was assumed to consist of shift workers based on the job nature but it seems that they are not necessarily night shift workers. 

“Results” section 

In my opinion it is repetitive and difficult to comprehend.  

The type of intervention in each study is very different (meal composition, healthier food, meal timing during shift work) and, it seems to me that the results are not comparable. The authors should do the differentiation during the disclosure of the effects of the dietary intervention. 

Minor comments 

“Introduction” section

Second paragraph (lines 37-51): Most of the data are from studies about shift work. The authors mention this point several times but it could be helpful to start the paragraph saying that it is what we know about shift work (no night shift)

Third paragraph (lines 52-65): I understand that the information is not directly extracted from studies in night shift workers. It seems to me that the authors review knowledge from facts that could be applied to this type of shift work. If it is correct, I think that the authors could start the paragraph stating the lack of direct data.

“Results” section

Line 220: The authors mention 3 programs but there are 4 citacions. “Blood pressure and/or heart rate were measured by 3 nutrition education programs [43,47,48,51]”

Reviewer 2 Report

Shift work, especially night shift has been associated with an elevated risk of chronic diseases. In this article, the authors nicely presented a concise literature review specifically focusing on the benefits of dietary interventions for night-shift workers.  However, a clear conclusion could not be reached due to huge variability in study designs, types of dietary interventions, and differences in outcome measures.

Comment1: The method section needs further clarification. For example, in the abstract authors mention a total of 758 articles were retrieved, but in method, the authors mention a total of 738 publications were found after excluding duplicates + 18 citations by reference searches totaling 756 articles. Similarly, Figure 1 says a total of 756 results were screened. Additionally, figure 1 is not clear. The steps are not self-explanatory and hard to relate back to the text in the method section.